# Saliva Has High Sensitivity and Specificity for Detecting SARS-CoV-2 Compared to Nasal Swabs but Exhibits Different Viral Dynamics from Days of Symptom Onset

**DOI:** 10.3390/diagnostics15151918

**Published:** 2025-07-30

**Authors:** Tor W. Jensen, Rebecca L. Smith, Joseph T. Walsh

**Affiliations:** 1Office of the Vice President for Economic Development and Initiatives, University of Illinois at Urbana-Champaign, Urbana, IL 61801, USA; jaywalsh@uillinois.edu; 2Cancer Center at Illinois, University of Illinois at Urbana-Champaign, Urbana, IL 61801, USA; 3Stephens Family Clinical Research Institute, Carle Foundation Hospital, Urbana, IL 61801, USA; 4Carle R. Woese Institute for Genomic Biology, University of Illinois at Urbana-Champaign, Urbana, IL 61801, USA; rlsdvm@illinois.edu; 5Department of Pathobiology, University of Illinois at Urbana-Champaign, Urbana, IL 61802, USA; 6Carle Illinois College of Medicine, University of Illinois at Urbana-Champaign, Urbana, IL 61801, USA; 7Department of Bioengineering, University of Illinois at Urbana-Champaign, Urbana, IL 61801, USA; 8Department of Biomedical Engineering, University of Illinois Chicago, Chicago, IL 60607, USA

**Keywords:** COVID-19, SARS-CoV-2, diagnostic, qPCR, saliva

## Abstract

**Background/Objectives**: Saliva as a diagnostic medium for COVID-19 requires fewer resources to collect and is more readily adopted across a range of testers. Our study compared an Emergency Use Authorized direct saliva-to-RT-qPCR test against an FDA-authorized nasal swab RT-qPCR assay for participants who reported symptoms of respiratory infection. **Methods**: We analyzed 737 symptomatic participants who self-selected to test at either a community testing facility or a walk-in clinic due to respiratory symptoms and provided matched saliva and nasal swab samples. Samples were collected between March and September of 2023, both before and after the declared end of the public health emergency. **Results**: A total of 120 participants tested positive in at least one of the tests. For participants testing in the first 5 days of reported symptoms, the saliva test had a 94.0 positive percent agreement (PPA; 95% C.I. 88.9–99.1%) with the nasal test and a 99.0 negative percent agreement (NPA; 95% C.I. 98.1–99.9%). The viral load decreased beyond day 1 of reported symptoms for saliva testing. Viral load increased up to day 4 for nasal swabs and then decreased. The same number of discordant positive samples (five each) occurred for both tests within 5 days of symptoms onset. **Conclusions**: In the endemic phase of COVID-19 and for development of new tests, testing methods that are less invasive are more likely to be adopted. The results of saliva-based versus nasal swab PCR measurements relative to days of symptom onset are needed to optimize future testing strategies.

## 1. Introduction

The World Health Organization (WHO) declared a Public Health Emergency of International Concern related to SARS-CoV-2 on 11 March 2020. More than three hundred molecular and antigen diagnostic tests or devices received Emergency Use Authorization in the United States for the detection of the virus in human samples. From March of 2020 to June of 2022, the United States averaged more than one million COVID-19 tests per day [1]. The broadly accepted gold standard of nasopharyngeal sampling (NPS) requires sampling swabs, containers, and viral transport media for collection as well as trained medical personnel to collect the sample. Anterior nasal sampling shows results comparable to NPS with less patient discomfort, but has similar sample collection supply chain needs, although with fewer medical personnel needs [2,3]. The volume of testing, particularly before testing production was scaled up, resulted in a scarcity of diagnostic reagents as well as materials used for sample collection. The shortage of sample collection materials and the high frequency of testing prompted the development of diagnostic tests for saliva sampling, which requires fewer collection materials and is less invasive to collect.

Prior to the COVID-19 pandemic, saliva had been proposed as an alternative to nasopharyngeal aspirate/swab or nasal swabs for the detection of flu and respiratory syncytial virus (RSV) [4,5,6,7]. Since the onset of the COVID-19 pandemic, saliva-based measurements have proven to be a reliable and accurate tool for diagnosing the presence of the virus [2,3,8,9,10,11,12,13,14,15,16,17,18,19,20,21,22]. At the time of this clinical study, the FDA website indicated more than 20 diagnostic assays had received Emergency Use Authorization for detection of SARS-CoV-2 in saliva. Commercial technology for saliva measurements relies primarily on targeted nucleic acid using reverse transcription, quantitative polymerase chain reaction (RT-qPCR; e.g., SalivaDirect, SalivaNow, covidSHIELD), end-point PCR with secondary detection (e.g., DxTerity), or loop-mediated isothermal amplification (LAMP; e.g., Metrix). Saliva sample pre-preprocessing can include nucleic acid extraction (e.g., Revvity), proteinaseK treatment (e.g., SalivaDirect, SalivaNow), and heat denaturation (e.g., SalivaDirect, SalivaNow, covidSHIELD). Testing programs developed by SalivaDirect and covidSHIELD can return test results to the participant within 24 h [14,23]. As of May 2024, the SalivaDirect (6.5 million [23]) and covidSHIELD (11 million) systems alone had performed more than 17.5 million tests, showing the potential of saliva-based measurements.

With the end of the federal COVID-19 public health emergency declaration in May of 2023, there has been a shift from public health monitoring of SARS-CoV-2 transmission to individual determination to test for the virus in order to make personal decisions on follow-up treatment [24]. With the expectation that SARS-CoV-2 infections will continue as an endemic disease, it is vital to include evaluation of saliva-based diagnostics for SARS-CoV-2 in the endemic phase. Saliva-based testing using an EUA-authorized test (EUA 202555) is compared against anterior nasal swabs and is evaluated in the context of days following symptom onset.

## 2. Materials and Methods

This prospective study was approved by the WIRB-Copernicus Group Institutional Review Board (tracking #20230721) and all participants provided informed consent.

### 2.1. Participants

This study enrolled individuals who self-elected to test for SARS-CoV-2 during the period of March to September of 2023, both before and after the end of the COVID-19 emergency declaration. The primary variant of SARS-CoV-2 during this period was Omicron. Community testing sites: Community SARS-CoV-2 testing sites were operated by the SHIELD Illinois group. SHIELD Illinois was a University of Illinois System initiative to provide the covidSHIELD saliva-based SARS-CoV-2 test developed by the University of Illinois to K–12 schools, colleges, universities, companies, and the public at sites across the state of Illinois. Participants were enrolled at 5 community testing sites in Illinois: the Vermillion County Health Department in Danville, the South Suburban Community College in South Holland, the Southern Illinois School of Medicine in Springfield, the PBCR in Rockford, and the Campus Recreation Center East in Urbana. Participation was offered to all presenting at these community testing sites by advertisements located in the testing site and by the personnel at the check in desk.

Participants were also enrolled at one clinical testing site: the McKinley Health Center on the campus of the University of Illinois, Urbana-Champaign. Patients visiting McKinley were informed of the study by clinic providers using flyers describing the study. The study was also advertised in the weekly student and staff informational email sent out by the University. The study site in McKinley was set up to match the general footprint used by community testing sites and was also operated by SHIELD Illinois employees.

### 2.2. Consent and Data Collection

At the testing desk, individuals were informed of the requirements of the study verbally or provided with a one-page description of the study prior to consenting. Inclusion criteria included the ability to consent for participants with respiratory symptoms who were 18 years or older or the consent of a parent or legal guardian for participants 17 years or younger. Participants were informed that they must be willing and able to provide saliva (drool) samples and be able to perform a nasal swab while observed by study personnel. Participants 17 years or younger could receive assistance with nasal swab from a parent or guardian under observation of study personnel. Lastly, the participants or their parent/legal guardian must be willing and able to answer questions about any symptoms. Participants unwilling or unable to provide consent or who had participated in the study in the previous 14 days were excluded from the study. Individuals who agreed to participate were consented using a secure REDCap electronic consenting system. Symptom data were captured via electronic survey and date of symptom onset was entered by the participant in REDCap using a pull-down calendar to select the first day of symptoms. The number of days since onset of symptoms was calculated based on the self-reported date of symptom onset and the date of testing; symptoms starting on the same day as testing were considered to be day 0, while those without symptoms were not included in the study. Participants self-reported symptoms from the list detailed in Table 1.

### 2.3. Sample Collection

After consent and symptom data collection, participants were instructed in the sample collection procedure per the instructions for both sample collection processes; in brief, they were as follows:1.Participants were provided with a preservative-free collection tube with funnel to collect 1–2 mL of saliva (drool). After collection, the participant removed the funnel and placed the cap on the collection vial, verified the tube number with study personnel, and placed the tube in a collection rack.2.Participants were then provided with a Roche cobas PCR Uni swab sample tube and instructed to collect a nasal swab sample by inserting the swab approximately 1 inch inside the nostril and then rubbing the swab in a circle 5 times for 10–15 s; this sample collection was repeated with the same swab inside the other nostril. The participant then placed the swab in the provided tube, snapped off the handle, placed the cap on the collection vial, verified the tube number with study personnel, and placed the tube in a collection rack.

The order of sample collection (first saliva, then nasal swab) was the same for all participants.

### 2.4. Saliva Testing

Saliva samples were handled per EUA (202555). Samples were transported from testing centers to central laboratory facilities in insulated containers at room temperature by covidSHIELD personnel or medical courier. Laboratory testing was completed within 48 h of collection. SARS-CoV-2 has been shown to be stable in raw saliva during this time [25]. Samples were tested for 3 SARS-CoV-2 specific genes (ORF, N, and S) per the covidSHIELD EUA protocol as detailed previously [14,26]. Samples were measured with the commercially available Thermo Fisher Scientific (Waltham, MA, USA) TaqPath COVID-19 Combo Kit (see catalog number A47814 for kit component details). This commercial kit has received Emergency Use Authorization from the FDA (EUA200010) for the detection of nucleic acid markers specific to SARS-CoV-2. All saliva samples were measured in SHIELD testing laboratories. Briefly, saliva samples were heated at 95 °C for 30 min followed by the addition of 2×x Tris/borate/EDTA/Tween20 buffer at a 1:1 ratio for a final Tween-20 concentration of 0.5%. Samples were then assayed using the Thermo Taqpath COVID-19 Combo kit. Samples with 2 of 3 target genes positive (Ct value < 40) were considered positive. Samples with indeterminate results were retested. After retesting, samples with indeterminate results were considered negative. Samples were determined invalid if the control gene did not return a valid result. Testing labs performing covidSHIELD testing were blinded to the nasal swab results.

### 2.5. Nasal Swab Testing

Nasal swab samples were frozen at −80 °C within 48 h of sample collection. Frozen samples were batched and assayed within 30 days of freezing with the FDA-authorized cobas SARS-CoV-2 Qualitative assay for use on the cobas 6800/8800 Systems (510(k) # K213804). The assay was run on a cobas 6800 system at the Carle Health Methodist Hospital following the manufacturer’s instructions. The Roche cobas assay tests for SARS-CoV-2 target and a pan-Sarbecovirus target. Samples testing positive for the SARS-CoV-2 target were considered positive. Samples testing negative for the SARS-CoV-2 target and positive for the pan-Sarbecovirus target were considered negative for SARS-CoV-2. Samples with indeterminate results were retested. After retesting, samples with indeterminate results were considered negative. Samples were determined invalid if the control gene did not return a valid result. The testing lab at Carle was blinded to the saliva testing results.

### 2.6. Data Analysis

When average Ct values for samples were calculated, only positive genes were included in the average. For nasal swab Ct values, when only the SARS-CoV-2 target was positive, that value was used. When either the saliva or nasal sample returned invalid results after retesting, both samples were removed from analysis. Kappa was calculated using the standard formula [27]. Confidence intervals for proportions were calculated with the binom.test function in R [28]. Statistical significance for differences in Ct values between tests was calculated using paired t-tests with the t.test function in the statistical package [28], while differences in Ct values by days since symptom onset were assessed using the emmeans package [29]. All statistical analyses were performed in Rstudio, and visualizations and data cleaning were created using tidyverse [30], with summaries created with tableone [31].

## 3. Results

A total of 59% of study participants were recruited at community testing sites, and 41% of study participants were recruited at the McKinley clinical site.

### 3.1. Demographics

Table 2 shows the demographic information for the study participants. The largest demographic group represented were non-Hispanic White (392/980; 40.0%), followed by Asian (210/980; 21.4%), and African American (208/980; 21.2%) participants. The majority of the participants reported as female (54.1%), followed by male (43.5%), nonbinary (1.4%), or did not answer (1.0%). The average age of participants was 34 years (SD 16 years) with a median age of 28. The youngest and oldest participants were 6 and 83 years old, respectively.

### 3.2. Testing Results

A total of 980 participants were enrolled in the study. To remove inducement to falsify symptoms, participants who did not report symptoms were allowed to provide samples but data from non-symptomatic participants were not used in the analysis. Four participants were either unwilling or unable to provide either saliva or a nasal swab sample, resulting in 976 participants providing samples. Due to labeling and shipping issues on two study days, seven saliva samples were lost or could not be processed. Five saliva samples returned invalid results and seven nasal samples returned invalid results. Therefore, 957 participants provided matched and valid saliva and nasal samples. Only participants with matched and valid samples were included in the analysis. Of the 980 participants in the study, 752 reported one or more symptoms. Of those reporting symptoms who provided matched samples, the average number of days following onset of symptoms was 4.4 days, with a median of 2 days. Reported days following onset of symptoms ranged from 0 to 91 days. The symptoms reported for those with valid test results are shown in Table 3.

A total of 737 participants both reported symptoms and had matched saliva and nasal swab testing results. Of those, 110 participants had a positive nasal result (14.9%) and 106 participants had a positive saliva result (14.4%). A total of 120 participants had a positive result for any test (16.3%). Of the 110 nasal positive participants, 96 tested positive by saliva, giving an overall positive predictive agreement (PPA) of 87.3%. In concordance with earlier studies with covidSHIELD [17], agreement between the saliva and nasal swab results was higher during earlier stages of infection. Table 4 shows the PPA and NPA by days of symptom onset, assuming that the nasal test is the gold standard. The negative predictive agreement between nasal and saliva results was 98.4% (617/627) for all days and greater than 98% for all subsets of symptomatic days. For those participants testing with 5 days or fewer symptoms, PPA was 94.0% (95% C.I. 88.9–99.1%) and NPA 99.0% (95% C.I. 98.1–99.9%). Kappa for the two tests overall was 0.88, indicating good agreement.

Presumed viral loads generally decreased, as indicated by increasing Ct values, with increasing days from symptom onset for saliva samples. For nasal swab samples, Ct values decreased from day 0 to day 4 of symptoms and increased after day 4 (Figure 1 and Table 5). Ct values in saliva were significantly higher than those in nasal swabs 2 to 5 days following symptom onset (Figure 1 and Table 5).

### 3.3. Discordant Samples

A total of 24 participants had discordant results between nasal swab and saliva, with 14 positive only on the nasal swab, and 10 positive only on saliva. Among participants with 5 or fewer days of symptoms, five participants had a positive nasal swab with a corresponding negative saliva, and an additional five participants had a positive saliva with a corresponding negative nasal swab (Table 6). The average Ct value for discordant nasal swabs was significantly higher than that of non-discordant swabs (*p* = 0.004), and the average Ct value for discordant saliva samples was significantly higher than that of non-discordant saliva samples (*p* = 0.007). There is no significant difference in average Ct values for discordant samples with 5 days of symptoms or less compared to those with 6 days or more of symptoms for either nasal swabs (*p* = 0.63) or saliva (*p* = 0.58).

## 4. Discussion

The context, implications, and actionable information related to COVID-19 testing has shifted massively from the onset of the pandemic to the current endemic phase of the disease. In the post-pandemic environment, responsibility for both testing and follow-up actions related to treatment and isolation has shifted to the individual and away from public health regulatory oversight [24]. As COVID-19 is likely to remain an endemic disease for the foreseeable future, the decision to test for the disease will continue to fall on the individual, with choices to either self-test or consult with a health care advisor to take a test. In this testing context, saliva and anterior nasal swabs are likely to be the most acceptable and easiest to obtain sample types for individuals who choose to test for the disease. In this large-scale study, we compare saliva-based COVID-19 testing to nasal swabs in real world community testing sites and walk-in clinical care settings for individuals who have chosen to test. As this study spanned the period just before and just after the end of the public health emergency declaration, these data may be reflective of conditions that will continue in the endemic phase of COVID-19. Thus, this study provides insight into the RT-qPCR sensitivity and viral loads in both saliva and anterior nasal swab samples relative to the day of symptom onset.

The data confirm results from previous studies that show saliva-based testing has high agreement with nasal swab testing during the first 5 days following symptom onset, with increasing discordance from day 6 onward [3,9,17,18,32,33,34]. During the first 5 days of symptoms, the number of discordant tests where the saliva sample is positive and the corresponding nasal sample is negative is the same as the number of discordant tests where the nasal sample is positive and the corresponding saliva sample is negative (i.e., five discordant tests in each case). This is similar to previous studies and begs the question about which compartment (nasal or oral) should be considered the true gold standard sample type for determining COVID-19 infection [3,15,16,35,36,37]. At 6 days of symptoms or greater, there are almost twice as many discordant nasal positives (9) than discordant saliva positives (5), which results in the declining positive predictive agreement for saliva testing the longer symptoms are present in participants. Regarding a positive COVID-19 test with symptoms and the need for isolation, as of December, 2023, the CDC guidelines indicate that “You are likely most infectious during these first 5 days”. Testing modalities that may be more reflective of infectious status might be more readily adopted by a population hesitant to have test results impact work or school schedules.

The testing targets (genes) and testing platforms used for saliva and nasal swab samples in this study are different. Therefore, signal strength comparisons between saliva and nasal swabs in matched samples are not feasible. There are no intra-day statistical differences in average Ct values for either test between any of days 0 to 4 of reported symptoms; however, trends in Ct values for each test method are instructive. For both saliva and nasal swab samples, the average Ct value at 7 days or more of symptoms is greater than that closer to symptom onset. For saliva samples, average Ct values reached a minimum 1 day following reported symptoms, with a gradual increase thereafter. For nasal swab samples, Ct values decreased up to 4 days following reported symptoms and then increased thereafter. The differences in the trends for the two compartments may reflect differences in production rates and turnover of fluids between oral saliva and nasal mucous [12,16,36,38,39,40,41,42,43], resulting in greater accumulation of viral particles in nasal mucus during the first 4 days of symptoms, or they may represent a difference in ongoing rates of viral particle production between the two compartments. Findings have shown that SARS-CoV-2 may have bacteriophagic behavior in both fecal and oral samples [44,45,46]. Overall health of the oral microbiome and viral loads in saliva may be indicators of infection susceptibility or disease severity, further emphasizing the diagnostic potential of saliva-based measurements of SARS-CoV-2 [45,47,48].

Compared to other diagnostic fluids, saliva has not been widely used as a diagnostic media, even though saliva has been successfully used for detection of COVID-19, Flu A and B, RSV, rhinovirus, enterovirus, metapneumovirus, and adenovirus, among others. In a review of publications from 1992 through 2022 by Laxton et al. [49], the authors note that 70% of publications exploring saliva as a diagnostic were published after 2019. It is not surprising that the interest in saliva as a diagnostic media gained traction in response to a rapidly evolving global pandemic. The covidSHIELD testing program used in these studies was developed by the University of Illinois [14] (EUA 202555), then used widely on campus, across Illinois, nationally, and globally. One major factor in the selection of saliva was the inability to source the necessary equipment to collect nasal swab samples or to perform large-scale RNA extraction procedures in the early days of the pandemic. Saliva can be collected in any sterile, non-reactive container and the protocol developed for covidSHIELD does not require additional reagents for RNA extraction. Because of this, covidSHIELD was successfully implemented using a community testing structure, allowing large numbers of samples to be collected at multiple sites with centralized lab facilities, resulting in test-to-result reporting in less than 24 h [14]. More than 10 million samples have been tested with covidSHIELD over the course of the pandemic.

## 5. Conclusions

The onset of the COVID-19 pandemic reinforced the strategic need to rapidly develop diagnostics to respond to emerging infectious diseases. Part of an efficient, rapid response is the optimized use of resources to avoid production and supply chain bottlenecks. During the endemic phase of the disease, ease of testing and ready test adoption are key to viable tests. The data presented support the continued use of saliva for diagnosing SARS-CoV-2 infections in the endemic phase of the disease. Due to the ease of sample collection relative to nasal swabs, the reduced resources and reagents required, and its diagnostic efficacy, saliva should be further explored as a diagnostic media for existing, and particularly for emerging, respiratory infections [3,8,11,13,14,20,50,51,52].

## Figures and Tables

**Figure 1 diagnostics-15-01918-f001:**
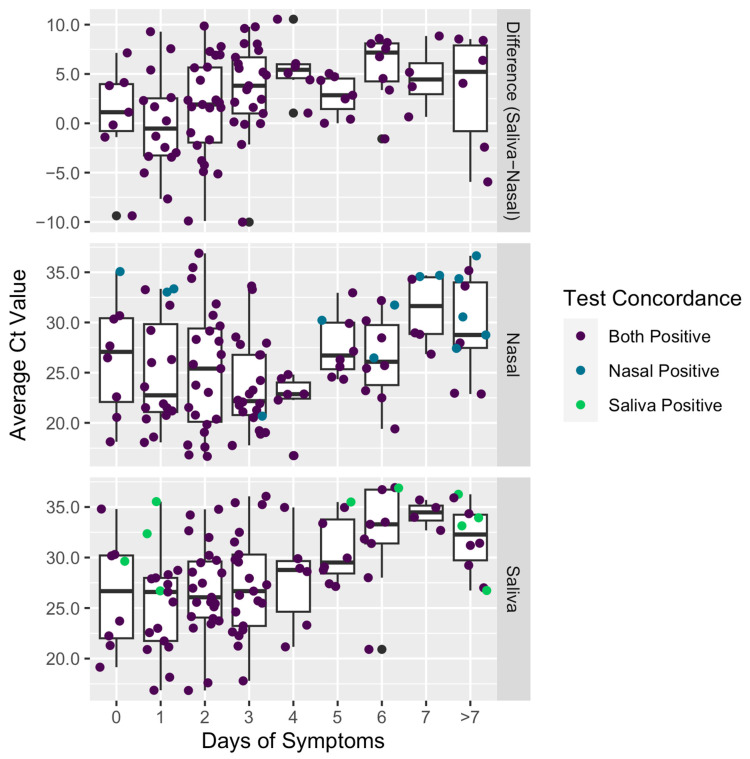
Average Ct values for nasal swab and saliva samples and the difference between the two values by number of symptomatic days prior to testing.

**Table 1 diagnostics-15-01918-t001:** Symptom report list.

	Symptom
1	Scratchy/painful sore throat
2	Painful sore throat
3	Cough (worse than baseline)
4	Runny nose
5	Symptoms of fever or chills
6	Temp > 100.4 °F or 38 °C
7	Muscle aches (greater than baseline)
8	Nausea, vomiting, diarrhea
9	Shortness of breath
10	Unable to taste or smell
11	Red or painful eyes
12	No symptoms

**Table 2 diagnostics-15-01918-t002:** Characteristics of participants in study compared to the results of SARS-CoV-2 RT-qPCR on saliva and nasal swabs.

Variable	Level	Both Negative	Both Positive	Nasal Positive	Saliva Positive
n		617	96	14	10
Gender (%)	Female	346 (56.1)	46 (47.9)	7 (50.0)	5 (50.0)
	Male	254 (41.2)	49 (51.0)	7 (50.0)	5 (50.0)
	Nonbinary	9 (1.5)	0 (0.0)	0 (0.0)	0 (0.0)
	Prefer not to answer	8 (1.3)	1 (1.0)	0 (0.0)	0 (0.0)
Race (%)	AIAN	8 (1.3)	0 (0.0)	0 (0.0)	0 (0.0)
	Asian	129 (21.8)	23 (24.2)	4 (30.8)	4 (40.0)
	Black	151 (25.5)	12 (12.6)	4 (30.8)	2 (20.0)
	Hispanic	104 (17.5)	15 (15.8)	1 (7.7)	0 (0.0)
	Pacific Islander/Hawaiian	1 (0.2)	0 (0.0)	0 (0.0)	0 (0.0)
	Self-described	1 (0.2)	0 (0.0)	0 (0.0)	0 (0.0)
	White	199 (33.6)	45 (47.4)	4 (30.8)	4 (40.0)
Age in years median [IQR]		26 [21, 39]	32 [21, 49]	23 [19, 30]	25 [22, 28]
Vaccinated (%)	No	30 (4.9)	2 (2.1)	1 (7.1)	1 (10.0)
	Yes	587 (95.1)	94 (97.9)	13 (92.9)	9 (90.0)
Days since first symptom, median [IQR]		2 [1, 4]	3 [2, 5]	6 [3, 8]	5 [1, 9]

**Table 3 diagnostics-15-01918-t003:** Symptoms reported by participants in this study comparing the results of SARS-CoV-2 RT-qPCR on saliva and nasal swabs.

	Both Negative	Both Positive	Nasal Positive	Saliva Positive
n	617	96	14	10
Scratchy painful sore throat (%)	290 (47.0)	64 (66.7)	6 (42.9)	4 (40.0)
Painful sore throat (%)	162 (26.3)	31 (32.3)	2 (14.3)	2 (20.0)
Cough worse than baseline (%)	236 (38.2)	69 (71.9)	7 (50.0)	4 (40.0)
Runny nose (%)	362 (58.7)	70 (72.9)	10 (71.4)	6 (60.0)
Symptoms of fever or chills (%)	84 (13.6)	42 (43.8)	7 (50.0)	0 (0.0)
Temp 100.4 °F or 38 °C (%)	13 (2.1)	11 (11.5)	1 (7.1)	0 (0.0)
Muscle aches greater than baseline (%)	119 (19.3)	39 (40.6)	4 (28.6)	1 (10.0)
Nausea, vomiting, diarrhea (%)	69 (11.2)	19 (19.8)	2 (14.3)	0 (0.0)
Shortness of breath (%)	73 (11.8)	16 (16.7)	2 (14.3)	2 (20.0)
Unable to taste or smell (%)	35 (5.7)	13 (13.5)	5 (35.7)	0 (0.0)
Red or painful eyes (%)	72 (11.7)	12 (12.5)	2 (14.3)	0 (0.0)

**Table 4 diagnostics-15-01918-t004:** Testing results by days of symptoms among participants this study comparing the results of SARS-CoV-2 RT-qPCR on saliva and nasal swabs.

Days of Symptoms	Both Negative	Both Positive	Nasal Positive	Saliva Positive	NPA (95% C.I.)	PPA (95% C.I.)
0	109	7	1	1	99.1% (95%, 100%)	87.5% (47.3%, 99.7%)
1	108	14	2	3	97.3% (92.3%, 99.4%)	87.5% (61.7%, 98.4%)
2	117	23	0	0	100% (96.9%, 100%)	100% (85.2%, 100%)
3	79	21	1	0	100% (95.4%, 100%)	95.5% (77.2%, 99.9%)
4	58	6	0	0	100% (93.8%, 100%)	100% (54.1%, 100%)
5	28	7	1	1	96.6% (82.2%, 99.9%)	87.5% (47.3%, 99.7%)
6	10	8	2	1	90.9% (58.7%, 99.8%)	80% (44.4%, 97.5%)
7	30	4	2	0	100% (88.4%, 100%)	66.7% (22.3%, 95.7%)

**Table 5 diagnostics-15-01918-t005:** Mean difference in average Ct values (saliva value minus nasal value) by days of symptoms for positive saliva and nasal swab samples in concordant samples.

Days Since Symptom Onset	Mean Difference in Ct Values	95% C.I.	*p*-Value
0	0.2	−2.62, 3.03	0.88
1	1.53	−0.64, 3.7	0.16
2	3.5	1.4, 5.59	0.0023
3	5.48	2.26, 8.71	0.0072
4	2.84	0.96, 4.71	0.01
5	5.69	2.78, 8.6	0.0024
6	4.6	−0.82, 10.01	0.074
7	3.17	−3.15, 9.49	0.25
>7	0.2	−2.62, 3.03	0.88

**Table 6 diagnostics-15-01918-t006:** The days of symptoms and Ct values for discordant samples with (A) positive nasal swab and negative saliva, and (B) positive saliva and negative nasal swab.

(A) Nasal Positive/Saliva Negative						
Days of Symptoms	Control		Covid		Pan Corona	
0	35.97		33.84		36.3	
1	33.77		32.8		33.9	
1	33.24		32.36		33.68	
3	35.1		20.64		20.7	
5	36.79		29.67		30.76	
6	35.64		31.25		32.22	
6	36.65		26.2		26.74	
7	33.63		34.99		34.15	
7	36.52		34.83		34.55	
8	36.32		28.24		29.27	
8	35.39		27.52		27.32	
8	34.88		34.4		34.33	
9	33.85		29.84		31.27	
10	34.79		37		36.28	
**(B)** **Saliva Positive/Nasal Negative**						
**Days of Symptoms**	**Control**	**ORF**		**N**		**S**
0	24.75373	29.48826		28.94565		30.46365
1	24.99574	32.80674		31.77399		32.48619
1	24.79328	26.82811		26.66201		26.60906
1	23.4059	36.11923		-		34.94921
5	24.46222	35.44045		-		35.55172
6	26.21524	-		36.69434		37.05559
8	24.2179	37.33555		35.18433		-
9	32.68468	27.74104		25.27331		27.18791
11	24.42438	33.9116		32.36166		-
15	24.01456	34.51339		32.75044		34.53242

## Data Availability

The raw data supporting the conclusions of this article will be made available by the authors on request.

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
