# Peer review of "Saliva Has High Sensitivity and Specificity for Detecting SARS-CoV-2 Compared to Nasal Swabs but Exhibits Different Viral Dynamics from Days of Symptom Onset"

_diagnostics, 2025, doi:10.3390/diagnostics15151918_

Round 1
Reviewer 1 Report
Comments and Suggestions for Authors
The study by Tor W. Jensen et asl. compares the reliability of salivary versus nasopharyngeal tests for COVID-19 infectiousness positivity. The study collects more than 700 specimens at the home district with proper approval of the local committee and performs statistical analysis reporting good promising results for the development of other salivary tests.
The study is well articulated but I only require some major revisions:
1. the authors need to extend the introduction and describe other salivary tests in the literature as well
2. the statistical analysis needs to be better articulated and the software used needs to be well introduced and explained
3. it is necessary to describe how salivary samples were transported
4. the tag sequences of the genes studied should be transcribed into a table and included in the paper
5. the authors should use the mdpi tamplate of the tables and format the tables according to the mdpi style
6. the kits used for which SARS-CoV-2 variant were manufactured. The collected samples were analyzed in the period of the Omicron variant and should be written in the paper
7. the use of salivary tests, as other studies have also seen fecal ones, should be integrated into the discussion of the new findings that SARS-CoV-2 also has betteriophagic behavior and analyzing their results in light of these studies places much more emphasis on the use of salivary tests.
Author Response
1. Summary |
|
|
Thank you very much for taking the time to review this manuscript. Please find the detailed responses below and the corresponding revisions/corrections highlighted/in track changes in the re-submitted files.
|
||
2. Questions for General Evaluation |
Reviewer’s Evaluation |
Response and Revisions |
Does the introduction provide sufficient background and include all relevant references? |
Can be improved |
|
Are all the cited references relevant to the research? |
Can be improved |
|
Is the research design appropriate? |
Can be improved |
|
Are the methods adequately described? |
Must be improved |
|
Are the results clearly presented? |
Must be improved |
|
Are the conclusions supported by the results? |
Can be improved |
|
Are all figures and tables clear and well-presented? |
Must be improved |
|
3. Point-by-point response to Comments and Suggestions for Authors |
||
Comments 1: the authors need to extend the introduction and describe other salivary tests in the literature as well |
||
Response 1: In addition to the saliva based testing research cited in the manuscript (see citations 2, 3, 8-22) we have added a review of the saliva testing technology used by several commercial platforms as below:
Since the onset of the COVID-19 pandemic, saliva-based measurements have proven to be a reliable and accurate tool for diagnosing the presence of the virus [2, 3, 8-22]. At the time of this clinical study, the FDA website indicated more than 20 diagnostic assays had received Emergency Use Authorization for detection of SARS-CoV-2 in saliva. Commercial technology for saliva measurements rely primarily on targeted nucleic acid using reverse transcription, quantitative polymerase chain reaction (RT-qPCR; e.g. SalivaDirect, SalivaNow, covidSHIELD), end-point PCR with secondary detection (e.g. DxTerity), or loop mediated isothermal amplification (LAMP; e.g. Metrix). Saliva sample pre-preprocessing can include nucleic acid extraction (e.g. Revvity), proteinaseK treatment (e.g. SalivaDirect, SalivaNow), and heat denaturation (e.g. SalivaDirect, SalivaNow, covidSHIELD). Testing programs developed by SalivaDirect and covidSHIELD can return test results to the participant within 24 hours [14, 23]. As of May 2024, the SalivaDirect (6.5 million [23]) and covidSHIELD (11 million) systems alone had performed more than 17.5 million tests, showing the potential of saliva-based measurements.
|
||
Comments 2: the statistical analysis needs to be better articulated and the software used needs to be well introduced and explained |
||
Response 2: The statistical software used (Rstudio) is well known in the field and is used with no modification in this applied study. The statistical analysis has been better articulated (see below). Text and references have been updated in the revised manuscript.
When average Ct values for samples was calculated, only positive genes were included in the average. For nasal swab Ct values, when only the SARS-CoV-2 target was positive, that value was used. When either the saliva or nasal sample returned invalid results after retesting, both samples were removed from analysis. Kappa was calculated using the standard formula [^]. Confidence intervals for proportions were calculated with the binom.test function in R.[25] Statistical significance for differences in Ct values between tests was calculated using paired t-tests with the t.test function in the stats package[*], while differences in Ct values by days since symptom onset were assessed using the emmeans package[26]. All statistics were performed in Rstudio, and visualizations and data cleaning were created using tidyverse[27], with summaries created with tableone.[28]
^ Cohen J. Weighted kappa: nominal scale agreement with provision for scaled disagreement or partial credit. Psychol Bull. 1968;70:213–220. * R Core Team (2023). _R: A Language and Environment for Statistical Computing_. R Foundation for Statistical Computing, Vienna, Austria. https://www.R-project.org/.
|
||
Comments 3: it is necessary to describe how salivary samples were transported |
||
Response 3: The text in Section 2.4 Saliva testing: “Saliva samples were handled per the EUA (202555) and sent to central facilities for testing within 48 hours of collection.” Has been expanded to describe sample transport: “Saliva samples were handled per the EUA (202555). Samples are transported from testing centers to central laboratory facilities in insulated containers at room temperature by covidSHIELD personnel or medical courier. Laboratory testing was completed within 48 hours of collection. SARS-CoV-2 has been shown to be stable in raw saliva during this time [24].”
|
||
Comments 4: the tag sequences of the genes studied should be transcribed into a table and included in the paper |
||
Response 4: The assay run on the saliva samples used a commercial kit from Thermo Fisher Scientific. The primer mix is proprietary to that kit and has received Emergency Use Authorization from the FDA. To further describe the assay, the text in Section 2.4 Saliva testing: “Samples were tested for 3 SARS-CoV-2 specific genes (ORF, N, and S) per the covidSHIELD EUA protocol as detailed previously.[14, 24]” Has been expanded to read “Samples were tested for 3 SARS-CoV-2 specific genes (ORF, N, and S) per the covidSHIELD EUA protocol as detailed previously.[14, 24] Samples were measured with the commercially available Thermo Fisher Scientific TaqPath COVID-19 Combo Kit (see catalog number A47814 for kit component details). This commercial kit has received Emergency Use Authorization from the FDA (EUA200010) for the detection of nucleic acid markers specific to SARS-CoV-2.”
|
||
Comments 5: the authors should use the mdpi template of the tables and format the tables according to the mdpi style |
||
Response 5: All tables have been formatted with the mdpi template and updated in the manuscript.
|
||
Comments 6: the kits used for which SARS-CoV-2 variant were manufactured. The collected samples were analyzed in the period of the Omicron variant and should be written in the paper |
||
Response 6: Agreed. We have included additional information on the test kit used for saliva testing (see comment 4 above). The Emergency Use Authorization for this kit was issued in May of 2022 after the emergence of the Omicron variant. To emphasize the time frame and relevant variant during the study the text in the manuscript “This study enrolled individuals who self-elected to test for SARS-CoV-2 during the period of March to September of 2023, both before and after the end of the COVID-19 emergency declaration.” Has been expanded to “This study enrolled individuals who self-elected to test for SARS-CoV-2 during the period of March to September of 2023, both before and after the end of the COVID-19 emergency declaration. The primary variant of SARS-CoV-2 during this period was Omicron.” Per suggestion from Reviewer 2, this text was moved from the Introduction Section to the Materials and Methods Section (2.1 Participants).
|
||
Comments 7: the use of salivary tests, as other studies have also seen fecal ones, should be integrated into the discussion of the new findings that SARS-CoV-2 also has betteriophagic behavior and analyzing their results in light of these studies places much more emphasis on the use of salivary tests. |
||
Response 7: We agree with the reviewer’s insight. The discussion has added the potential impact of this mechanism on the diagnostic utility of saliva measurements.
“The differences in the trends for the two compartments may reflect differences in production rates and turnover of fluids between oral saliva and nasal mucous[12, 16, 34, 36-41] resulting in greater accumulation of viral particles in nasal mucus during the first 4 days of symptoms, or they may represent a difference in ongoing rates of viral particle production between the two compartments. Findings have shown that SARS-CoV-2 may have bacteriophagic behavior in both fecal and oral samples.[42-44] Overall health of the oral microbiome and viral loads in saliva may be indicators of infection susceptibility or disease severity, further emphasizing the diagnostic potential of saliva-based measurements of SARS-CoV-2.[43, 45, 46]”
|
Reviewer 2 Report
Comments and Suggestions for Authors
The authors performed a comparative analysis of saliva and nasal swab samples for detecting SARS-CoV-2 infection in a symptomatic group of individuals enrolled at five community testing sites in Illinois, USA (737 participants, of whom 120 tested positive by at least one of the methods).
They found good concordance between the results obtained from the two sample types during the first five days following symptom onset, with increasing discordance observed from day six onward. The comparable diagnostic sensitivity and specificity of the two tests is notable, as saliva collection is easy to perform, more comfortable than nasopharyngeal swabbing, and may therefore improve patient compliance and facilitate large-scale screening and surveillance efforts.
While the topic is not entirely new—given the growing interest in recent years in the use of saliva for diagnosing various respiratory virus infections—this study may help further stimulate discussion around viral load distribution in the nose and mouth, as well as the differing dynamics of SARS-CoV-2 compartmentalization.
Overall, the paper is well-written, the data are clearly presented, and the discussion is supported by up-to-date references.
I suggest the following minor revisions:
- Page 2, lines 66–70: It would be more appropriate to move this sentence to the Materials and Methods section.
- Page 2, lines 70–73: Consider moving this sentence to the Discussion or Conclusions section to avoid prematurely revealing the study’s main findings.
- Please verify page 4, line 168.
- The term “Ct values” should be defined in the Abbreviations section.
- In the caption for Table 6, the word “values” is repeated; also, the distinctions A) and B) within the legend should be clarified.
Author Response
1. Summary |
|
|
Thank you very much for taking the time to review this manuscript. Please find the detailed responses below and the corresponding revisions/corrections highlighted/in track changes in the re-submitted files.
|
||
2. Questions for General Evaluation |
Reviewer’s Evaluation |
Response and Revisions |
Does the introduction provide sufficient background and include all relevant references? |
Yes |
|
Are all the cited references relevant to the research? |
Yes |
|
Is the research design appropriate? |
Yes |
|
Are the methods adequately described? |
Yes |
|
Are the results clearly presented? |
Yes |
|
Are the conclusions supported by the results? |
Yes |
|
3. Point-by-point response to Comments and Suggestions for Authors |
||
Comments 1: Page 2, lines 66–70: It would be more appropriate to move this sentence to the Materials and Methods section. |
||
Response 1: Agreed. We have moved those lines to the Materials and Methods section.
|
||
Comments 2: Page 2, lines 70–73: Consider moving this sentence to the Discussion or Conclusions section to avoid prematurely revealing the study’s main findings. |
||
Response 2: Agreed. We have moved those lines to the Conclusions section.
|
||
Comments 3: Please verify page 4, line 168. |
||
Response 3: That information is correct. The statistical analysis has also been better articulated (see below). Text and references have been updated in the revised manuscript.
When average Ct values for samples was calculated, only positive genes were included in the average. For nasal swab Ct values, when only the SARS-CoV-2 target was positive, that value was used. When either the saliva or nasal sample returned invalid results after retesting, both samples were removed from analysis. Kappa was calculated using the standard formula [^]. Confidence intervals for proportions were calculated with the binom.test function in R.[25] Statistical significance for differences in Ct values between tests was calculated using paired t-tests with the t.test function in the stats package[*], while differences in Ct values by days since symptom onset were assessed using the emmeans package[26]. All statistics were performed in Rstudio, and visualizations and data cleaning were created using tidyverse[27], with summaries created with tableone.[28]
^ Cohen J. Weighted kappa: nominal scale agreement with provision for scaled disagreement or partial credit. Psychol Bull. 1968;70:213–220. * R Core Team (2023). _R: A Language and Environment for Statistical Computing_. R Foundation for Statistical Computing, Vienna, Austria. https://www.R-project.org/.
|
||
Comments 4: The term “Ct values” should be defined in the Abbreviations section. |
||
Response 4: We have added Ct value to the Abbreviations section (“Cycle threshold value when target is detected”).
|
||
Comments 5: In the caption for Table 6, the word “values” is repeated; also, the distinctions A) and B) within the legend should be clarified. |
||
Response 5: We apologize to the reviewer for our error. The caption for Table 6 was mistakenly copied from Table 5. The correct caption for Table 6 has been updated to “The days of symptoms and Ct values for discordant samples with A) positive nasal swab and negative saliva and B) Positive saliva and negative nasal swab.”
|
Round 2
Reviewer 1 Report
Comments and Suggestions for Authors
The authors have solved my concern.